# Knowledge, Attitudes, and Risk Perception of Broiler Grow-Out Farmers on Antimicrobial Use and Resistance in Oyo State, Nigeria

**DOI:** 10.3390/antibiotics11050567

**Published:** 2022-04-24

**Authors:** Nurudeen O. Oloso, Ismail A. Odetokun, Ibraheem Ghali-Mohammed, Folorunso O. Fasina, Isaac Olufemi Olatoye, Victoria O. Adetunji

**Affiliations:** 1Nuafus (Agroveterinary) Ventures, 14 Olalekan Alabi, Abayomi Street, Off Abayomi Bus Stop, Iwo Road, Ibadan 200263, Nigeria; 2Department of Veterinary Public Health and Preventive Medicine, University of Ilorin, Ilorin 240272, Nigeria; ghali.mi@unilorin.edu.ng; 3Emergency Centre for Transboundary Animal Diseases-Food and Agriculture Organisation of the United Nations (ECTAD-FAO), Dar es Salaam 14111, Tanzania & Department of Veterinary Tropical Diseases, University of Pretoria, Onderstepoort 0110, South Africa; folorunso.fasina@fao.org; 4Department of Veterinary Public Health and Preventive Medicine, University of Ibadan, Ibadan 200284, Nigeria; femi.olatoye@ui.edu.ng (I.O.O.); voadetunji@mail.ui.edu.ng (V.O.A.)

**Keywords:** antimicrobial use, antimicrobial resistance (AMR), knowledge, attitude, risk perception, broiler grow-out farmers, Nigeria

## Abstract

Assessing knowledge, attitudes, and risk perception of Nigerian broiler grow-out farmers (*n* = 152) to antimicrobial resistance (AMR) with a five sectional purposive-structured-questionnaire: demographics; knowledge; attitudes; risk-perception; and response to regulation of antimicrobial practices. Data were analyzed using descriptive statistics, chi-square test, and binary logistic regression. Respondents’ knowledge score, in total, was lower than average, with all (100%) respondents having the understanding that antibiotics kill/reduce bacteria, most participants (>73%) believing that feeding antibiotics to broiler chickens is a necessity for weight gain, and many (>69%) thinking that no negative side-effects exist with the use of antibiotics. Poor attitude towards antimicrobial usage was prevalent (>63%) with unsatisfactory performance in most instruments: >60% of farmers reported using antimicrobials every week and still use antimicrobials when birds appear sick, and most (>84%) arbitrarily increase the drug dosages when used. However, a satisfactory performance score was reported (68%) in risk perception of AMR with >63% perceiving that inappropriate use of antibiotics is the main factor causing the emergence of resistant bacteria; >65.8% expressed that AMR in broiler chickens is not essential for public health, that AMR cannot develop from broiler bacteria diseases, that increasing the frequency of antimicrobial use cannot increase AMR in future, and that usage cannot lead to antibiotic residue in broiler-meat products leading to AMR development in human. None of the respondents were aware of any regulation for monitoring antimicrobial use. Significant factors associated with knowledge, attitudes, and risk perception of antimicrobial use and resistance among broiler grow-out farmers include marital status, farm category, education, educational specialization, sales target, growth duration/cycle, broiler stocking batch, and feed source. Identified gaps exist in AMR awareness among Nigerian broiler farmers and should be targeted through stakeholders’ participation in combatting AMR threats.

## 1. Introduction

Antimicrobial resistance (AMR) is a growing public health threat of broad concern to countries and multiple sectors [1,2,3]. Governments worldwide continue to pay attention to it as a threat to modern medicine [1]. The emergence and spread of drug-resistant pathogens that have acquired new resistance mechanisms, leading to AMR, threaten our ability to treat common infections [1,2,3]. Especially alarming is the rapid global spread of multi- and pan-resistant bacteria (also known as “superbugs”) that cause infections that are not treatable with commonly existing antimicrobials such as antibiotics [1,2,3].

Previous successes that followed the invention of antimicrobials are being challenged rapidly by AMR [3]. Every new antimicrobial developed has been subsequently reported to be challenged by AMR. The clinical pipeline of new antimicrobials is becoming dry [3,4,5]. In recent times, the World Health Organization (WHO) identified 32 antibiotics in clinical development that address the WHO list of priority pathogens in 2019, of which only six were classified as innovative [1,2,3,5]. The WHO further observed a lack of access to quality antimicrobials as a major issue in many countries, with antimicrobial shortages affecting countries at all levels of development, agriculture, and environment, especially in the healthcare systems of developing countries due to the sociopolitical economic situation [1,4].

In the developing world, particularly Africa, these consequences are underreported due to inadequate research, epidemiological, and surveillance data [1,2,3,6,7]. AMR has been projected to cause about 4.15 million human deaths per annum by 2050 if left to continue without concerted control of this menace in Africa [6]. In most developing countries, such as Nigeria, emphasis on AMR control is mainly placed on incidences in humans, with very little attention paid to animals (the livestock, wildlife, and poultry industries, as well as veterinary clinical practices) [8]. Discrepancies exist, and the level of attention to AMR among the public, animal, and environmental health sectors differ. Hence resource allocations to control AMR in Nigeria have wide gaps with unintended consequences leading to threatened food security amidst the AMR challenge [8,9].

Antimicrobials are increasingly becoming ineffective as AMR spreads worldwide, with infections becoming more challenging to treat, leading to death and increasing mortality rates [2]. Therefore, there is an urgent need for new antibacterials, for example, for treatment of carbapenem-resistant gram-negative bacterial infections which are identified in the WHO list of priority pathogens [1,5,6]. However, strong awareness must be awakened among concerned individuals so that new antimicrobial development would not suffer the same fate as the currently ineffective ones [1,2,3,6].

The cost of AMR to national economies and their health systems is high as it affects the productivity of patients or their caretakers through prolonged hospital stays and the need for more expensive and intensive care in humans and animals [1,2,5,6,7].

Without practical tools for the prevention and adequate treatment of drug-resistant infections and improved access to existing and new quality-assured antimicrobials, the number of people for whom treatment is failing or who die of diseases will increase [3]. Medical procedures, such as surgery, including caesarean sections or hip replacements, cancer chemotherapy, and organ transplantation, will become riskier [1,2,3].

AMR occurs naturally, but misuse of antibiotics in humans and animals is accelerating the process. AMR is undermining many other advances in medicine. Tackling AMR is a high priority globally coordinated by the WHO and OIE under the United Nations [1,2,3]. The global campaign aims to raise awareness towards encouraging best practices among the public, policymakers, health, and agriculture professionals [2,5,6,8]. Situation analysis of AMR in Nigeria revealed a very high level and trend of AMR in humans, animals, and the environment [8,9]. The Nigerian broiler production value chain has shown evidence of overreliance on and indiscriminate use of antimicrobial drugs and detection of multi-drug resistant pathogens (NBPVC) [10]. Scarce resources prevail on awareness of AMR as linked with antimicrobial use in NBPVC [10]. With the increase in broiler production encouraged by the human population increase, there is the need for animal protein to be produced within the shortest possible period and to serve economic interests. Therefore, there is a need to monitor the awareness and use of antimicrobials among grow-out broiler farmers on the chicken produced for human consumption [10,11].

Therefore, this study was designed to explore the level of awareness about AMR in antimicrobial use and resistance among broiler grower farmers through the study of knowledge, attitudes to practices, and risk perception of broiler grow-out farmers in Oyo State of Nigeria.

## 2. Material and Methods

### 2.1. Structure of Target Population

Broiler grow-out farmers in Oyo State of Nigeria were the target population comprising the owners, the managers, and supervisors of poultry farms where broilers are raised regularly from day-old chicks to target size and weight. The population targeted were those involved in the decision pattern of antimicrobial administration during broiler grow-out production. The target population among the study population comprised veterinarians, para-veterinarians, broiler poultry farmers, poultry supervisors, other farm workers, and non-veterinary-oriented workers. Oyo State was selected as a preferred representative poultry production hub for Nigeria based on the consensus of 76% of all respondents reported in a previous study, with reasons including the presence of the only central open day-old chicks market in the country, the main point for several stakeholders, the highest number of head offices or presence of industrial poultry companies, the highest number of hatcheries and poultry abattoirs [12]. Furthermore, two out of the four companies licensed by the Federal Republic of Nigeria (FRN) to import grandparent stock (GS) have their GS farms in Oyo State. Between September 2016 and September 2017, members of the Poultry Association of Nigeria (PAN), Oyo State, and the Nigerian Veterinary Medical Association (NVMA), Oyo State branch were interviewed (*n* = 464) [10,12]. Oyo State, Nigeria, is located between geographical coordinates 8.1196° N, 3.4196° E (Figure 1). Based on the information obtained, we identified six stages of the value chain as the most critical in the NBPVC, including the following: breeder, hatchery, grow-out, abattoir, retail, and live-bird-market [10,12].

### 2.2. Study Design, Sample Size, and Sampling Protocol

A cross-sectional questionnaire—Knowledge, Attitudes, and Practices (KAP) survey—was carried out among grow-out farmers in Oyo State of Nigeria through a face-to-face farm visitation done when biological samples were collected concurrently for some other studies from August 2016 to April 2017 [10]. KAP surveys represent a specific population to collect information on what is known, believed, and done concerning a particular topic and are the most frequently used study tool in health-seeking behavior research [13]. The KAP survey is a quantitative method (predefined questions formatted in standardized questionnaires) that provides access to quantitative and qualitative information [13,14]. KAP surveys reveal misconceptions or misunderstandings that may represent obstacles to the activities we would like to implement and potential barriers to behavior change [13,14]. A KAP survey essentially records an “opinion” and is based on the “declarative” (i.e., statements). In other words, the KAP survey reveals what was said, but there may be considerable gaps between what is said and what is done [13]. A KAP survey can measure the extent of a known situation, confirm or disprove a hypothesis, and provide new tangents of a situation’s reality [13]; enhance the knowledge, attitudes, and practices of specific themes; identify what is known and done about various health-related subjects [13,14]; establish the baseline (reference value) for use in future assessments and help measure the effectiveness and ability of health education activities to change health-related behaviors [6,13]. Furthermore, such surveys suggest an intervention strategy that reflects specific local circumstances and the cultural factors that influence them while planning activities suited to the respective population involved [13].

The sample size formula for a cross-sectional study (random sample) [15,16,17] was used. The assumptions used to calculate the sample size were the percentage of respondents with an expected level of knowledge and awareness set at 90%, the absolute precision was at 95% degree of confidence, and a level of error of 5%. A sample size of 138 was arrived at with the calculation computed using OpenEpi [16,17]. To make up for non-response, 20% contingency was added. Therefore, a minimum of 175 respondents were targeted for data collection. Purposive sampling was carried out so that at least 25 respondents were recruited in each of the six geopolitical zones of Oyo state.

### 2.3. Questionnaire Design, Implementation, and Data Collection

A structured questionnaire comprised five sections: (1) demographics; (2) knowledge level of antimicrobials; (3) attitudes to antimicrobial use; (4) risk perception of antimicrobial link to AMR; and (5) practices related to antimicrobial use in response to regulation. We conducted a literature search including the standard recommended instrument of global bodies [1,2,13]. The questionnaire was translated into the “Yoruba” language to accommodate farmers who are native speakers. The survey tool was pre-tested on a group of poultry farmers. The list of broiler farmers was collected from PAN, screened to select those active in operation, and claimed that they would be in operation for the following year. The questionnaire was administered to the targets with work experience ≥1 year. Seven enumerators were involved during the questionnaire administration.

The demographic questions included: age of respondent, marital status, age of farm, years of experience as broiler farmer, farm category, educational level of the respondent, areas of educational specialization, sales target purpose, growth duration per batch of each cycle of broiler, broiler stocking capacity per batch, and feed source. We believe asking those demographic questions will expose the personal potential to have an adequate knowledge level to understand antimicrobial use and AMR contribution.

To assess the general antimicrobial knowledge level of the participants, we discussed and asked several questions from each of them that were summed up in nine direct questions: antibiotics are necessary for broiler chickens for weight gain, antibiotics don’t kill bacteria, antibiotics are painkillers, antibiotics are antipyretic, all antibiotics show the same curative effect, antibiotics cannot be harmful to beneficial bacteria in the broiler gut, antibiotics are effective on other organisms, antibiotics are effective on ecto- and endoparasites, and antibiotics have no side effects.

To assess the attitudes of grow-out broiler farmers toward the practice of antibiotic use in Oyo State, we evaluated the response of each participant to 10 questions: “it is not necessary to consult a veterinarian before using antibiotics in broiler”, “I use antibiotics every week during the production cycle”, “I use antibiotics immediately when birds get sick”, “I get information from other farmers and sources other than veterinarians”, “I increase the dose of antibiotics if the response is not satisfactory”, “I increase the frequency of antibiotics if the response is not satisfactory”, “I don’t fully read to understand the information on the label and prospectus before usage”, “I stop giving antibiotics during treatment if the birds feel better even if it is after a day”, “I rely more on the recommendations of other farmers and sources of the birds even if a veterinarian is not involved”, and “I only consulted veterinarian when the birds get sick and fail to respond to treatments attempted”.

To assess the risk perception to AMR by broiler grower farmers, we evaluated the response of each participant using 10 questions on already reported contributory factors to the risk of the development of AMR, exposing how these participants perceive the risk of AMR formation in the broiler chickens being raised for human consumption.

We assessed antibiotics use practices and responses to regulations with six open-ended questions and reported based on direct categorization. The questions involved where they store antimicrobials on the farm, the number of days used to treat broilers, how they handle leftover antibiotics after each cycle of broiler raised, how long excess antibiotics are stored for reuse, frequency of administering antibiotics to each batch of broiler grown, and level of awareness through exposure to training or understanding on any of the listed topics related to antibiotics use in animals, antibiotic use linked to antimicrobial resistance, and antibiotic residue in food.

The questionnaire was pre-tested on five farmers, and the responses were used to adjust the questions before a final questionnaire was adopted. Feedback received was used to standardize the questions before administration to respondents. Informed consent from each participant was obtained despite receiving formal approval from PAN, Oyo State chapter. Respondents were allowed to participate in the survey voluntarily (not in a group setting) and had the opportunity to withdraw from participation without bias based on recommendations of the 2013 World Medical Association Declaration of Helsinki ethical principles for medical research involving human subjects [17]. The protocol and procedure employed in the study were ethically reviewed and approved before the commencement of the study. We obtained approval from the Animal Ethics Committee of the University of Pretoria, South Africa (V062-15 on 1 August 2015) and the Ministry of Health, Oyo State of Nigeria (ref: AD13/479/433). In addition, consent for this research was sought from PAN through the association’s chapter in Oyo State; the same organization encouraged farmers’ participation in the study. However, none of the individuals or organizations influenced the study design or implementation of the project.

### 2.4. Data Management and Statistical Analysis

Data collected were summarized in Microsoft Office Excel 2013 and analyzed using SPSS version 20 and the Open Source Epidemiologic Statistics for Public Health (OpenEpi), version 3.03a [16]. The Cronbach’s α, a numerical reliability coefficient, was determined as 0.922 for the questions asked [18,19]. The data were presented using descriptive statistics and percentages. Three outcome variables (Table 1) were developed: (1) knowledge level; (2) attitudes to the practice of antimicrobial usage; (3) risk perception of AMR. A numeric scoring system [20,21,22,23] was computed from a five-point Likert scale [24] to assess these outcome variables. A reverse scoring pattern was used where strongly disagree was the highest correct answer, scored as five. The outcome variables were further expressed as either satisfactory or unsatisfactory based on a cut-off point (i.e., satisfactory scores = scores > mean + 1 standard deviation) (Table 1). The demographic characteristics were used as the independent variables for evaluating the associations with satisfactory knowledge, attitudes, and risk perception of AMR by the respondents. Chi-square test (including Fisher’s exact test for binary variables) was used to test for the association. Additionally, a binary logistic regression analysis was used to identify independent variables serving as underlining factors to good knowledge, attitudes, and risk perception of AMR. All statistical analyses were performed with a *p*-value < 0.05 considered significant.

## 3. Results

### 3.1. Demographic Information

A total of 152 participants (Age: 28 to 48 years) across Oyo State of Nigeria completed the questionnaire. Table 2 gives the demographic characteristics of all the respondents. The majority of respondents were male (73.7%), married (51.2%), had post-secondary school education (80.3%), and had exposure to agriculture/veterinary-oriented tertiary education (56.6%).

### 3.2. Knowledge Level of Antimicrobials in Broiler Grower Farmers

The level of knowledge of broiler grow-out farmers about AMR is average, as observed by their responses to the knowledge questions (Table 3). General knowledge scores were average but high in some instruments, such as all (100.0%) knowing that antibiotics kill or reduce bacteria. General knowledge ranged from 24 to 43, with a total score of 45 with a mean score of 32.7 ± 4.6 (Table 1). Overall, these farmers (50.0%) demonstrated average satisfactory knowledge scores on antimicrobials. Many still believe that antibiotics are necessary for broiler chickens for weight gain, with 34.2% strongly agreeing and 39.5% agreeing. None of them had a firm agreement with the statements: antibiotics don’t kill bacteria, antibiotics are painkillers, antibiotics are antipyretic, all antibiotics show the same curative effect, and antibiotics are effective on ecto- and endoparasites (Table 3).

### 3.3. Attitudes of Grow-Out Broiler Farmers to the Practice of Antibiotic Use in Oyo State

Respondents to this survey demonstrated general unsatisfactory and poor attitudes to antimicrobial use (Table 1). All questions showed poor and very poor attitudes to the use of antimicrobials. A moderate to high number of the respondents strongly agree (15.8%) and agree (31.6%) that it is unnecessary for them to consult a veterinarian before using antibiotics in growing their broilers (Table 4). Thereby 13.2% and 47.4% of the farmers strongly agree and agree, respectively, that they use antibiotics regularly and every week during each production cycle without any recourse to veterinary consultation. Despite the weekly antimicrobial usage, most of them start using antibiotics immediately when they feel the birds are sick (strongly agree 21.1%; agree 39.0%), and over 80% of the participants increase the dose and frequency of antibiotics if they feel the treatment of their sick birds is unsatisfactory after initial treatments. Yet, most of the respondents were in strong agreement (22.4%) and 39.5% in agreement that they do not thoroughly read to understand the information on the label and prospectus of the drugs before usage because they (21.1% and 42.1% of the respondents) rely more on the recommendations of other farmers and sources of the birds even if a veterinarian is not involved. Despite this attitudinal treatment pattern, 14.5% strongly agree, and 36.0% agree that they stop giving antibiotics during treatment if the birds feel better, even after a day (Table 4). The respondents (strongly agree: 15.8% and agree: 46.1%) only consulted veterinarians when the birds got sick and failed to respond to treatments attempted (Table 4).

### 3.4. Risk Perception to Antimicrobial Resistance by Broiler Grower Farmers

With regards to the key terms and question statements used to measure risk perception on the development of AMR among broiler grower farmers in their products (broiler birds) and a link to humans, the awareness level and risk perception among respondents appeared satisfactorily high in general (Table 1) and in most of the instruments (Table 5). Over 63% demonstrated satisfactory risk perception of the contribution of inappropriate use of antibiotics as the main factor causing the emergence of resistant bacteria, with 40.1% disagreeing and 22.4% strongly disagreeing with the instrument question, “It is not true that inappropriate use of antibiotics is the main factor causing the emergence of resistant bacteria”. This pattern appeared the same in most of the instruments with: antibiotics resistance in broilers is not essential for public health (43.4% disagreed, 22.4% strongly disagreed); bacteria causing disease in broilers cannot become resistant to antibiotics (39.5% disagreed, 26.3% strongly disagreed); increase in frequency of antimicrobial use cannot increase potentials of the resistance effects in future (38.2% disagreed, 28.9% strongly disagreed); use of antibiotics in broiler cannot lead to antibiotic residue in broiler meat products (44.7% disagreed, 22.4% strongly disagreed); antibiotic residue in broiler meat products cannot cause antibiotic resistance development in humans consuming them (38.2% disagreed, 25.0% strongly disagreed); antimicrobial use in broilers does not affect me, my family, and the public directly or indirectly (38.2% disagreed, 27.6% strongly disagreed). However, they believe that control of antimicrobial use in growing broilers will lead to more damages than benefits, with over 60% of the participants agreeing to the instrument question, “Restriction of antimicrobial use in growing broiler will lead to more damages than benefits” (17.1% strongly agreed, 43.4% agreed). It was noted that the majority of the respondents strongly agreed (9.2%) and agreed (50.0%) with the instrument, “If I know unconscious use of antimicrobials in broilers will give any harm to public health, I would continue to use antibiotics in broilers if my products will not be rejected”. This response becomes more critical with the split willingness of respondents to use antimicrobials for present value alone without consideration for the future.

### 3.5. Practice of Antibiotics Use and Response to Regulation

None of the participants is aware of any specific regulation or law in Nigeria controlling antimicrobial usage in broiler production; therefore, this is not reflected in the result. However, they are aware of the recommendation on the label of every antimicrobial package. Yet, there is a generally very poor response in following the regulation as found in the drug label recommendations. The majority do not have a medicine cabinet for medications and refrigerators for specific biologicals. Many (68.0%) of the respondents store antibiotics in any part of the poultry house (over 53%), and 14.5% store them in any other place anywhere on the farm, with just a few respondents (18.4%) keeping antibiotics in the medicine cabinet.

As against the claim of the majority that they source information from other sources outside the veterinarian (Table 4), some (40.8%) claimed they, however, follow the advice of a veterinarian on the number of days to administer antibiotics in each antibiotic regimen (Table 6) while few (11.8%) of them follow label instructions. This is also close to the claim of following the advice of a veterinarian (44.0%) on the frequency of antibiotics for each batch of broiler, with 11.8% of respondents using it every week or when the birds appear sick, whereas others (23.7%) use it according to a scheduled timetable or based on the condition of the birds.

Concerning training and exposure to improve participants’ awareness of AMR issues, most of them (above 80%) had attended at least one training session on the listed topics: antibiotics use in animals, antibiotic use and antimicrobial resistance, and antibiotic residue in food. More than half of them claimed to have received training in all the listed topics, demonstrating high awareness about antimicrobial resistance.

### 3.6. Demographic Factors Influencing the Knowledge, Attitudes, and Risk Perception of AMR among Broiler Grow-Out Farmers

Marital status, farm category, education, educational specialization, sales target, growth duration/cycle, broiler stocking batch, and feed source were demographic variables significantly associated with knowledge levels of antimicrobial use and resistance among broiler grow-out farmers in Oyo State, Nigeria (Table 7). Industrial broiler producers (OR = 17.3; 95% CI: 5.59–53.40; *p* < 0.001) were significantly more likely to possess adequate knowledge than commercial producers. Respondents with tertiary/post-secondary education level (OR = 11.5; 95% CI: 0.59–222.70; *p* < 0.126) and those with agriculture/veterinary-oriented post-secondary educational specialization (OR = 9.0; 95% CI: 2.89–28.13; *p* < 0.001) were more likely to have satisfactory knowledge of antimicrobial use and resistance. Furthermore, participants reporting higher broiler stocking/batch, e.g., 20,001 and above (OR = 43.2; 95% CI: 12.51–149.20; *p* < 0.001), possessed significantly more satisfactory knowledge of AMR than those reporting less than 5000 broiler stock. However, broiler producers of 40–56 days (OR = 0.2; 95% CI: 0.07–0.54; *p* = 0.001) and >56 days (OR = 0.2; 95% CI: 0.07–0.73; *p* = 0.018) growth duration/cycle were significantly less likely to possess satisfactory knowledge levels of antimicrobial use and resistance than producers of <40 days growth duration/cycle.

Table 8 presents the factors affecting attitudes to antimicrobial use practices and resistance among broiler grow-out farmers in Oyo State, Nigeria. Marital status, age category, experience as a broiler farmer, farm category, education, educational specialization, sales target, growth duration/cycle, broiler stocking/batch, and feed source were demographic variables significantly associated with attitudes to the practice of the respondents on AMR. The factors are almost similar to those observed with the respondents’ satisfactory knowledge levels on AMR, with the addition of the age category and experience as broiler farmer variables. Respondents with higher age category levels were less likely to have satisfactory attitudes to the practice of antimicrobial use and resistance. Participants with 25–35 years of experience as broiler farmers (OR = 0.0; 95% CI: 0.00–0.98; *p* = 0.009) were significantly less likely to demonstrate satisfactory attitudes toward the practice of antimicrobial use and resistance than farmers with 1–10 years of experience on the job. The factors influencing the risk perception to antimicrobial use and resistance among broiler grow-out farmers in Oyo State, Nigeria, are shown in Table 9. As observed in the factors on respondents’ attitudes to the practice of AMR, marital status, experience as broiler farmer, farm category, education, educational specialization, sales target, growth duration/cycle, broiler stocking/batch, and feed source were associated factors influencing the risk perception of the surveyed grow-out farmers to antimicrobial use and resistance.

## 4. Discussion

This study investigated and presented the level and associated factors of knowledge, attitudes to use, and risk perception of AMR of farmers in the Nigerian broiler grow-out setting. Our search revealed this study as the first that focused on exploring the contribution of the farmers’ broiler production setting to the awareness of AMR threat. Therefore, there are scarce resources available to compare and relate the demographics of respondents in this region to other areas globally. Overall evaluation exposed average knowledge level scores and poor attitudes to antimicrobial usage, but awareness and risk perception are satisfactorily high (Table 1, Table 2, Table 3, Table 4, Table 5 and Table 6). We found that certain independent factors (marital status, age category, experience as broiler farmer, farm category, education, educational specialization, sales target, growth duration/cycle, broiler stocking/batch, and feed source) influenced the levels of the observed knowledge, attitudes, and risk perception of AMR. This is in partial corroboration with studies regarding risk perceptions of poultry farmers in Cameroon and Nigeria, but those studies were without specific focus on broiler settings [11,25]. The survey of poultry farmers in Cameroon showed an overall low mean knowledge score on antimicrobial use and AMR, desirable attitude, appropriate practice towards antimicrobial use, and positive risk perception of AMR [25]. In comparison, the survey of Nigerian poultry farmers in Kwara State demonstrated that antimicrobial resistance knowledge is high, with satisfactory high-risk perception but a poor attitude to antimicrobial usage [11].

The average antimicrobial knowledge score in this study is below expectation because the majority of the participants possess post-secondary school education (80.0%), and the majority specialized in agriculture/veterinary-oriented fields (56.0%) as farm owners, managers, or supervisors. This, however, is reflected in their awareness of participation in training or awareness programs where AMR-related topics were discussed. It is, however, disappointing that there is such a poor attitude toward practices of antimicrobial usage. This study demonstrated that the motives and approaches are based mainly on the immediate economic value and response to the antimicrobials they currently use, without considering future implications. The majority (over 73%) of the respondents still believe in the old knowledge that antibiotics are necessary for broiler production despite understanding that antibiotics have side effects. The survey participants rely on information from other farmers and sources of day-old broiler chickens, and the practice is also encouraged by the labeling on some antibiotics. Another study in another part of Africa found that commonly used antibiotics were often labeled for prophylactic, growth promotion, and egg production improvement purposes in Kenya [26]. Considering the short life span of broiler birds from farm to table, poor attitudinal use of antimicrobials, and the driving factors for antimicrobial use, there is a heightened fear of high antimicrobial residue in broiler meat being offered for human consumption. A study also found that knowledge, attitudes, and practices significantly varied across five African countries, with poultry farmers demonstrating more knowledge, desirable attitudes, and prudent practices than pastoralist households [27]. That study showed that variation in knowledge, attitudes, and practices is related to several factors, including gender, disease dynamics on the farm, and source of animal health information [27], some of which were similarly observed in this study. Therefore, interventions to limit AMR should be based upon a bottom-up understanding of antimicrobial use at the farm level given limited inputs from animal health professionals and under-resourced regulatory capacities within most low- and middle-income countries [27].

Compared with studies on other production systems in Nigeria, poor knowledge of antimicrobial use and resistance has been demonstrated by farmers raising commercial and local poultry birds in Nigeria. Several pathways and factors facilitating the emergence of AMR were found [28]. Furthermore, unsatisfactory knowledge and practices on antimicrobial use and resistance were found among local dairy farmers [29]. This is also the situation among small ruminant farmers in Nigerian rural livestock communities [30]. Poor attitudes to antimicrobial use and resistance were also reported among freshwater fish farmers [31], pig producers [32], and cattle-rearers [33] in various parts of Nigeria.

None of the participants was aware of any specific regulation or law controlling antimicrobial usage in broiler production in Nigeria. However, they are aware of the recommendations on the label of every antimicrobial package. Even so, there is a generally poor response in following the regulation as found in the drug label recommendations. The majority of the respondents did not have a medicine cabinet for medications or refrigerators for specific biologics. Farmers’ access to these drugs in Nigeria is uncontrolled as they easily purchase the drugs over-the-counter without prescriptions from qualified veterinarians in various shops and markets [12]. This is one of the primary drivers of AMR in the country [28,29,30,31] and a critical point to control the widespread AMR challenges. Concerning training and exposure to improve awareness of participants to issues relating to AMR, most of the broiler farmers (above 80%) had attended at least one training session on the topics to improve awareness and risk perception of AMR. Such topics they claim awareness on are antibiotic use in animals, antibiotic use and its link with AMR, and antibiotic residue in foods linked to AMR. More than half of the respondents claimed to have received training in all the listed topics demonstrating there is high awareness about AMR. The success of regulation and monitoring has been reported in Sweden [34], where veterinarians worked closely with farmers, and farmers felt involved in the development of animal health management methods. The One Health concept was well-known among stakeholders at the national level but not at the farm level. Close cooperation between stakeholders seems to facilitate the development of animal production with low use of antibiotics [16].

Some limitations were associated with this study, especially with the use of a questionnaire. However, we ensured that the questionnaire was pre-tested with relevant questions asked and specific information obtained from the targeted participants [35]. During the questionnaire administration, we deliberately asked specific questions in ways that would confirm the consistency in the responses to the questionnaire provided by the respondents [35].

## 5. Conclusions

This study expounded the unsatisfactory knowledge level and attitudes with associated factors on AMR issues existing among broiler grow-out farmers in Nigeria despite the satisfactory perception of AMR threats. However, unwillingness to improve was observed because farmers are concerned about economic survival impulse, with little consideration for the future. Identified knowledge, attitudes, and risk perception gaps of awareness of AMR in Nigeria’s broiler industry should be targeted by the government through stakeholders’ participation in combatting AMR issues in Nigeria.

## Figures and Tables

**Figure 1 antibiotics-11-00567-f001:**
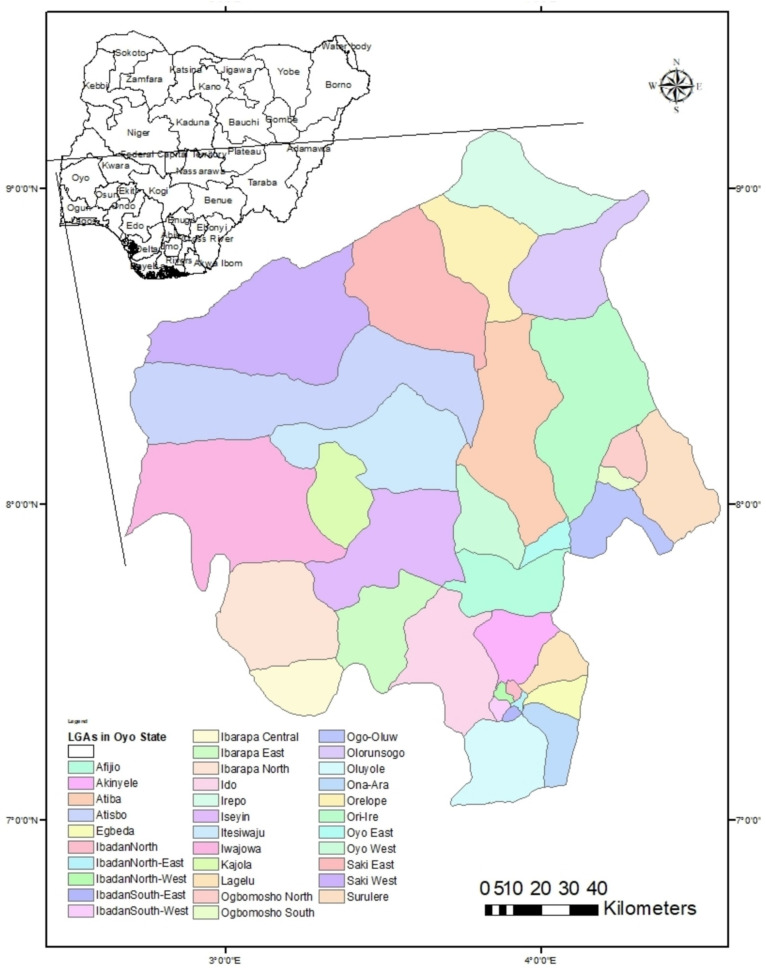
Map of Nigeria with the study area showing sampled local government.

**Table 1 antibiotics-11-00567-t001:** Description of scores from outcome variables of respondents.

Outcome Variable	Maximum Obtainable Score	Scores Obtained by Respondents		Mean ± SD	Unsatisfactory *n* (%)	^1^ Satisfactory *n* (%)
		Lowest	Highest			
Knowledge level of antimicrobials	45	24	43	32.7 ± 4.6	76 (50.0%)	76 (50.0%)
Attitudes to practices of antimicrobial usage	50	14	46	25.9 ± 10.5	96 (63.2%)	56 (36.8%)
Risk perception of AMR	50	16	46	33.6 ± 9.1	48 (31.6%)	104 (68.4%)

Note: ^1^ Satisfactory scores = scores > mean + 1 standard deviation. SD = Standard deviation. AMR = Antimicrobial resistance.

**Table 2 antibiotics-11-00567-t002:** Demographic characteristics of respondents.

Variables	*n* (%)	Variables	*n* (%)
Gender Male Female	112 (73.7) 40 (26.3)	Education Primary Secondary Post-secondary	4 (2.6) 26 (17.1) 122 (80.3)
Marital Status Married Single Separated	78 (51.3) 52 (34.2) 22 (14.5)	Educational specialization Secondary Non-agric. post-secondary Agric./vet-oriented post-secondary	30 (19.7) 36 (23.7) 86 (56.6
Age category (years) <28 28–32 33–37 38–42 43–47 >47	22 (14.5) 36 (23.7) 40 (26.3) 20 (13.2) 12 (7.9) 22 (14.5)	Sales target Non-contractual Contractual Company-owned	28 (19.7) 98 (64.5) 32 (21.1)
Farm age category (years) 1–10 11–15 16–25 26–35	28 (18.4) 36 (23.7) 64 (42.1) 24 (15.8)	Growth duration/cycle <40 days 40–56 days >56 days	110 (72.4) 26 (17.1) 16 (10.5)
Experience as broiler farmer (years) 1–10 11–15 16–25 26–35	82 (53.9) 40 (26.3) 26 (17.1) 4 (2.6)	Broiler stocking/batch 100–3000 3001–10,000 12,000–25,000 30,000–50,000	44 (28.9) 44 (28.9) 54 (35.5) 10 (6.6)
Farm category Traditional Commercial Industrial	32 (21.1) 48 (31.6) 72 (47.4)	Feed source Self-milling Commercial feed milling Finished feed	56 (36.8) 24 (15.8) 72 (47.4)

**Table 3 antibiotics-11-00567-t003:** Knowledge level of antimicrobial in broiler grower farmers.

The Questions Leading to Knowledge Outcome Variables	Strongly Agree	Agree	Neutral	Disagree	Strongly Disagree
Antibiotics are necessary for broiler chickens for weight gain	34.2%	39.5%	2.6%	18.4%	5.3%
Antibiotics don’t kill bacteria	0.0%	0.0%	0.0%	60.5%	39.5%
Antibiotics are painkillers	0.0%	19.7%	31.6%	36.8%	11.8%
Antibiotics are antipyretic	0.0%	32.9%	14.5%	36.8%	15.8%
All antibiotics show the same curative effect	0.0%	11.8%	9.2%	51.3%	27.6%
Antibiotics cannot be harmful to beneficial bacteria in the broiler gut	14.5%	18.4%	23.7%	27.6%	15.8%
Antibiotics are effective on other organisms	1.3%	6.6%	2.6%	23.7%	65.8%
Antibiotics are effective on ectoparasites and endoparasites	0.0%	5.3%	5.3%	68.4%	21.0%
Antibiotics have no side effects	1.3%	7.9%	21.1%	51.3%	18.4%

**Table 4 antibiotics-11-00567-t004:** Attitude of grow-out broiler farmers to the practice of antibiotic use in Oyo State.

Questions Leading to Attitude Variables	Strongly Agree	Agree	Neutral	Disagree	Strongly Disagree
Not necessary to consult a veterinarian before using antibiotics in broilers	15.8%	31.6%	9.2%	30.3%	13.2%
I use antibiotics every week during the production cycle	13.2%	47.4%	2.6%	17.1%	19.7%
I use antibiotics immediately when birds get sick	21.1%	39.5%	0.0%	17.1%	22.4%
I get information from other farmers and sources other than veterinarians	18.4%	47.4%	3.9%	25.0%	5.3%
I increase the dose of antibiotics if response is not satisfactory	38.2%	46.1%	1.3%	6.6%	7.9%
I increase the frequency of antibiotics if response is not satisfactory	34.2%	48.7%	1.3%	9.2%	6.6%
I don’t thoroughly read to understand the information on the drug label and prospectus before usage	22.4%	39.5%	11.8%	17.1%	9.2%
I stop giving antibiotics during treatment if the birds feel better, even if it is after a day	14.5%	36.8%	7.9%	25.0%	15.8%
I rely more on the recommendations of other farmers and sources of the birds, even if a veterinarian is not involved	21.1%	42.1%	1.3%	19.7%	15.8%
I only consulted veterinarians when the birds got sick and failed to respond to treatment attempted	15.8%	46.1%	1.3%	21.1%	15.8%

**Table 5 antibiotics-11-00567-t005:** Risk perception to antimicrobial use and resistance by broiler grower farmers.

Questions	Strongly Agree	Agree	Neutral	Disagree	Strongly Disagree
It is not true that inappropriate use of antibiotics is the main factor causing the emergence of resistant bacteria	10.5%	14.5%	11.8%	40.8%	22.4%
Antibiotic resistance in broilers is not important for public health	5.3%	19.7%	9.2%	43.4%	22.4%
Bacteria causing diseases in broilers cannot become resistant to antibiotics	3.9%	17.1%	13.2%	39.5%	26.3%
An increase in the frequency of antimicrobial use cannot increase the potential of the resistance effects in future	7.9%	15.8%	9.2%	38.2%	28.9%
Use of antibiotics in broilers cannot lead to antibiotic residues in broiler meat products	9.2%	14.5%	9.2%	44.7%	22.4%
Antibiotic residues in broiler meat products cannot cause antibiotic resistance development in humans consuming them	1.3%	22.4%	13.2%	38.2%	25.0%
Antimicrobial use in broilers does not affect me, my family, and the public indirectly	3.9%	21.1%	9.2%	38.2%	27.6%
Restriction of antimicrobial use in growing broilers will lead to more damages than benefits	17.1%	43.4%	13.2%	18.4%	7.9%
If I know that the unconscious use of antimicrobials in broilers will cause any harm to public health, I would continue to use antibiotics in broilers if my products will not be rejected	9.2%	50.0%	9.2%	21.1%	10.5%
If I know that the antibiotics I used may not work in the future, I will still not reduce their use if I think they will work presently	18.4%	23.7%	9.2%	35.5%	13.2%

**Table 6 antibiotics-11-00567-t006:** Practice of antibiotics use in response to regulation (*n* = 152%).

Where do you store the antibiotics? *n* (%)								
Medicine cabinet			Any part of the poultry house		Refrigerator		Just any other place	
28 (18.4)			82 (53.9)		20 (13.2)		22 (14.5)	
How many days do you use antibiotics to treat broiler? *n* (%)								
1 day	2 days	3 days	4 days	5 days	7 days	As directed by a veterinarian	As directed on the label	When symptoms stop
2 (1.3)	2 (1.3)	20 (13.2)	4 (2.6)	16 (10.5)	10 (6.6)	62 (40.8)	18 (11.8)	18 (11.8)
How would you handle the residual or leftover antibiotics? *n* (%)								
For the treatment of other sick birds				Another batch of broiler			Dispose	
24 (15.8)				54 (35.5)			74 (48.7)	
How long do you store the residual antibiotics for reuse (month)? *n* (%)								
1 month		3 months		7 months	<12 months		>12 months	I don’t store
2 (1.3)		6 (3.9)		18 (11.8)	24 (15.8)		28 (18.4)	74 (48.7)
How frequently do you give antibiotics to each batch of broiler you grow? *n* (%)								
Every week		As I feel the birds need them		As scheduled		When the birds are sick	As recommended by a veterinarian	
18 (11.8)		14 (9.2)		36 (23.7)		18 (11.8)	66 (43.4)	
Did you receive any training or awareness on the listed subjects? *n* (%)								
Antibiotics use in animals		Antibiotic use and antimicrobial resistance		Antimicrobial resistance	Antibiotic residue in food		No training at all	Received training in all listed subjects
14 (9.2)		4 (2.6)		12 (7.9)	10 (6.6)		30 (19.7)	82 (53.9)

**Table 7 antibiotics-11-00567-t007:** Factors influencing knowledge levels of antimicrobial use and resistance among broiler grow-out farmers in Oyo State, Nigeria.

Variables	Unsatisfactory *n* (%)	Satisfactory *n* (%)	Odds Ratio (OR)	95% Confidence Interval (CI)	*p*-Value
Gender					
Male	58 (51.8)	54 (48.2)	-		
Female	18 (45.0)	22 (55.0)	-	-	-
Marital status					
Married	50 (64.1)	28 (35.9)	1		
Single	22 (42.3)	30 (57.7)	0.4	0.20, 0.84	0.023 *
Separated	4 (18.2)	18 (81.8)	8.0	2.47, 26.10	<0.001 *
Age category (years)					
<28	8 (36.4)	14 (63.6)	-		
28–32	18 (50.0)	18 (50.0)	-	-	-
33–37	22 (55.0)	18 (45.0)	-	-	-
38–42	14 (70.0)	6 (30.0)	-	-	-
43–47	4 (33.3)	8 (66.7)	-	-	-
>47	10 (45.5)	12 (54.6)	-	-	-
Farm age category (years)					
1–10	18 (64.3)	10 (35.7)	-		
11–20	38 (52.8)	34 (47.2)	-	-	-
21 and above	20 (38.5)	32 (61.5)	-	-	-
Experience as broiler farmer (years)					
1–10	46 (56.1)	36 (43.9)	-		
11–15	18 (45.0)	22 (55.0)	-	-	-
16–25	10 (38.5)	16 (61.5)	-	-	-
25–35	2 (50.0)	2 (50.0)	-	-	-
Farm category					
Commercial	44 (61.1)	28 (38.9)	1		
Industrial	4 (8.3)	44 (91.7)	17.3	5.59, 53.40	<0.001 *
Traditional	28 (87.5)	4 (12.5)	0.2	0.07, 0.71	0.010 *
Education					
Primary	4 (100.0)	0 (0.0)	1		
High/Secondary	22 (84.6)	4 (15.4)	1.5	0.06, 33.17	>0.999
Tertiary/Post-secondary	50 (40.9)	72 (59.0)	11.5	0.59, 222.70	0.126
Educational specialization					
Secondary school	26 (86.7)	4 (13.3)	1		
Non-agric.-oriented post-secondary	14 (38.9)	22 (61.1)	10.2	2.93, 35.57	<0.001 *
Agric/Vet.-oriented post-secondary	36 (41.9)	50 (58.1)	9.0	2.89, 28.13	<0.001 *
Sales target					
Non-contractual	16 (72.7)	6 (27.3)	1		
Contractual	58 (59.2)	40 (40.8)	1.8	0.66, 5.11	0.349
Company-owned	2 (6.3)	30 (93.8)	40.0	7.23, 221.50	<0.001 *
Growth duration/cycle					
<40 days	44 (40.0)	66 (60.0)	1		
40–56 days	20 (76.9)	6 (23.1)	0.2	0.07, 0.54	0.001 *
>56 days	12 (75.0)	4 (25.0)	0.2	0.07, 0.73	0.018 *
Broiler stocking/batch					
100–5000	54 (84.4)	10 (15.6)	1		
5001–10,000	8 (33.3)	16 (66.7)	10.8	3.65, 31.94	<0.001 *
10,001–20,000	10 (35.7)	18 (64.3)	9.7	3.48, 27.12	<0.001 *
20,001 and above	4 (11.1)	32 (88.9)	43.2	12.51, 149.20	<0.001 *
Feed source					
Self-compounding and milling	8 (14.3)	48 (85.7)	1		
Self-compounding milled at a feed mill	18 (75.0)	6 (25.0)	0.1	0.02, 0.18	<0.001 *
Finished commercial feeds	50 (69.4)	22 (30.6)	0.1	0.03, 0.18	<0.001 *

*—significant at *p* < 0.05.

**Table 8 antibiotics-11-00567-t008:** Factors affecting attitudes to practices of antimicrobial use and resistance among broiler grow-out farmers in Oyo State, Nigeria.

Variables	Unsatisfactory *n* (%)	Satisfactory *n* (%)	Odds Ratio (OR)	95% Confidence Interval (CI)	*p*-Value
Gender					
Male	38 (33.9)	74 (66.1)	-		
Female	10 (25.0)	30 (75.0)	-	-	-
Marital status					
Married	36 (46.2)	42 (53.8)	1		
Single	8 (15.4)	44 (84.6)	4.7	1.97, 11.31	<0.001 *
Separated	4 (18.2)	18 (81.8)	3.9	1.19, 12.44	0.029 *
Age category (years)					
<28	0 (0.0)	22 (100.0)	1		
28–32	14 (38.9)	22 (61.1)	0.0	0.00, 0.69	<0.001 *
33–37	12 (30.0)	28 (70.0)	0.0	0.00, 0.68	<0.001 *
38–42	10 (50.0)	10 (50.0)	0.0	0.00, 0.47	<0.001 *
43–47	2 (16.7)	10 (83.3)	0.1	0.00, 2.94	0.159
>47	10 (45.5)	12 (54.5)	0.0	0.00, 0.56	<0.001 *
Farm age category (years)					
1–10	12 (42.9)	16 (57.1)	-		
11–20	20 (27.8)	52 (72.2)	-	-	-
21 and above	16 (30.8)	36 (69.2)	-	-	-
Experience as broiler farmer (years)					
1–10	22 (26.8)	60 (73.2)	1		
11–15	16 (40.0)	24 (60.0)	0.6	0.25, 1.22	0.207
16–25	6 (23.1)	20 (76.9)	1.2	0.4342, 3.44	0.919
25–35	4 (100.0)	0 (0.0)	0.0	0.00, 0.98	0.009 *
Farm category					
Commercial	20 (27.8)	52 (72.2)	1		
Industrial	4 (8.3)	44 (91.7)	4.2	1.345, 13.31	0.014 *
Traditional	24 (75.0)	8 (25.0)	0.1	0.05, 0.33	<0.001 *
Education					
Primary	4 (100.0)	0 (0.0)	1		
High/Secondary	24 (92.3)	2 (7.7)	0.8	0.02, 29.18	>0.999
Tertiary/Post-secondary	20 (16.4)	102 (83.6)	51.0	1.91, 1362.00	0.004 *
Educational specialization					
Secondary school	28 (93.3)	2 (6.7)	1		
Non-agric.-oriented post-secondary	10 (27.8)	26 (72.2)	36.4	7.28, 182.00	<0.001 *
Agric/Vet.-oriented post-secondary	10 (11.6)	76 (88.4)	106.4	21.94, 515.90	<0.001 *
Sales target					
Non-contractual	12 (54.6)	10 (45.4)	1		
Contractual	30 (30.6)	68 (69.4)	2.7	1.06, 6.98	0.064
Company-owned	6 (18.7)	26 (81.3)	5.2	1.53, 17.64	0.014 *
Growth duration/cycle					
<40 days	20 (18.2)	90 (81.8)	1		
40–56 days	20 (76.9)	6 (23.1)	0.1	0.02, 0.19	<0.001 *
>56 days	8 (50.0)	8 (50.0)	0.2	0.07, 0.66	0.017 *
Broiler stocking/batch					
100–5000	36 (56.3)	28 (43.7)	1		
5001–10,000	4 (16.7)	20 (83.3)	6.4	1.97, 20.95	0.002 *
10,001–20,000	4 (14.3)	24 (85.7)	7.7	2.39, 24.81	<0.001 *
20,001 and above	4 (11.1)	32 (88.9)	10.3	3.25, 32.51	<0.001 *
Feed source					
Self-compounding and milling	6 (10.7)	50 (89.3)	1		
Self-compounding milled at a feed mill	16 (66.7)	8 (33.3)	0.1	0.02, 0.19	<0.001 *
Finished commercial feeds	26 (36.1)	46 (63.9)	0.2	0.08, 0.56	0.001 *

*—significant at *p* < 0.05.

**Table 9 antibiotics-11-00567-t009:** Factors influencing risk perception to antimicrobial use and resistance among broiler grow-out farmers in Oyo State, Nigeria.

Variables	Unsatisfactory *n* (%)	Satisfactory *n* (%)	Odds Ratio (OR)	95% Confidence Interval (CI)	*p*-Value
Gender					
Male	72 (64.3)	40 (35.7)	-		
Female	24 (60.0)	16 (40.0)	-	-	-
Marital status					
Married	58 (74.4)	20 (25.64)	1		
Single	28 (53.9)	24 (46.1)	2.5	1.18, 5.24	0.026 *
Separated	10 (45.5)	12 (54.5)	3.5	1.31, 9.28	0.024 *
Age category (years)					
<28	12 (54.6)	10 (45.4)	-		
28–32	20 (55.6)	16 (44.4)	-	-	-
33–37	24 (60.0)	16 (40.0)	-	-	-
38–42	16 (80.0)	4 (20.0)	-	-	-
43–47	6 (50.0)	6 (50.0)	-	-	-
>47	18 (81.8)	4 (18.2)	-	-	-
Farm age category (years)					
1–10	18 (64.3)	10 (35.7)	-		
11–20	46 (63.9)	26 (36.1)	-	-	-
21 and above	32 (61.5)	20 (38.5)	-	-	-
Experience as broiler farmer (years)					
1–10	42 (51.22)	40 (48.8)	1		
11–15	28 (70.0)	12 (30.0)	0.5	0.20, 1.00	0.074
16–25	22 (84.6)	4 (15.4)	0.2	0.06, 0.60	0.004 *
25–35	4 (100.0)	0 (0.0)	0.1	0.00, 2.79	0.116
Farm category					
Commercial	70 (97.2)	2 (2.8)	1		
Industrial	0 (0.0)	48 (100.0)	4200.0	138.20, 127,700.00	<0.001 *
Traditional	26 (81.25)	6 (18.75)	8.1	1.53, 42.58	0.019 *
Education					
Primary	4 (100.00	0 (0.00)	1		
High/Secondary	22 (84.6)	4 (15.4)	1.8	0.06, 55.59	>0.999
Tertiary/Post-secondary	70 (57.4)	52 (42.6)	7.4	0.28, 195.40	0.344
Educational specialization					
Secondary school	26 (86.7)	4 (13.3)	1		
Non-agric.-oriented post-secondary	22 (61.1)	14 (38.9)	4.1	1.19, 14.41	0.038 *
Agric/Vet.-oriented post-secondary	48 (55.8)	38 (44.2)	5.2	1.65, 16.02	0.003 *
Sales target					
Non-contractual	18 (81.8)	4 (18.2)	1		
Contractual	72 (73.5)	26 (26.5)	1.6	0.50, 5.25	0.602
Company-owned	6 (18.8)	26 (81.2)	19.5	4.81, 79.12	<0.001 *
Growth duration/cycle					
<40 days	60 (54.6)	50 (45.4)	1		
40–56 days	24 (92.3)	2 (7.7)	0.1	0.02, 0.44	<0.001 *
>56 days	12 (75.0)	4 (25.0)	0.4	0.12, 1.32	0.199
Broiler stocking/batch					
100–5000	58 (90.6)	6 (9.4)	1		
5001–10,000	22 (91.7)	2 (8.3)	0.9	0.16, 4.69	>0.999
10,001–20,000	12 (42.9)	16 (57.1)	12.9	4.18, 39.72	<0.001 *
20,001 and above	4 (11.1)	32 (88.9)	77.3	20.32, 294.40	<0.001 *
Feed source					
Self-compounding and milling	12 (21.4)	44 (78.6)	1		
Self-compounding milled at a feed mill	20 (83.3)	4 (16.7)	0.1	0.02, 0.19	<0.001 *
Finished commercial feeds	64 (88.9)	8 (11.1)	0.0	0.01, 0.09	<0.001 *

*—significant at *p* < 0.05.

## Data Availability

Not applicable.

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
