# Peer review of "Knowledge, Attitudes, and Risk Perception of Broiler Grow-Out Farmers on Antimicrobial Use and Resistance in Oyo State, Nigeria"

_antibiotics, 2022, doi:10.3390/antibiotics11050567_

Round 1

Reviewer 1 Report

It is very positive of authors to address the AMR situation in Oyo State, Nigeria. Studies from all over the globe are most welcome. Questioning farmers and stakeholders is a socio-economic approach. The study is based on experience or impressions from this questioned group from Oyo State.

I am not sure why regularly stakeholders are mentioned in the article. I think it concerns 152 broiler farmers.

The biggest concern about this article is the fact that it is presented as a descriptive study only in numbers of participants and percentages. In itself that is a good start, but it is not an in depth-analysis. Antibiotics is a high impact journal requiring a more sophisticated analysis, in my opinion. A second layer of analysis and suitable methods would be expected to be performed to associate the responses to underlying factors as size of farm, i.e. stocking batch and age category of farmer and farm category. Because numbers are rather small, you may make logical categories (add categories together). For instance, I would not be surprised that Industrial farms score quite different from other categories. The same with age groups.

Introduction is the best part.

Table 1 in Material and Methods 2.4 is confusing.  I think it are results. First 3 columns: which units? Maximum obtainable score? It is clear what is a satisfactory score. I guess the unsatisfactory score are just all other respondents' answers. I can imagine the Demographic characteristics (table 2) to belong to the Material and methods section.

I would also like to see the presented demograhic data from this studied sample to be compared with the overall situation of the sector in that State. In other words, how does the sample compare to the total broiler farming sector characteristic in this State (and Nigeria).

Discussion part should preferably refer to more published research about attitudes towards AMR, like in the pig, veal calves and cattle sectors to gain a wider perspective of the results.

Author Response

Reviewers’ comments

Reviewer #1

It is very positive of authors to address the AMR situation in Oyo State, Nigeria. Studies from all over the globe are most welcome. Questioning farmers and stakeholders is a socio-economic approach. The study is based on experience or impressions from this questioned group from Oyo State.

Response: We thank the reviewer for the time and efforts inculcated to provide a positive criticism of this manuscript. We found the comments passed very useful and they have been used to improve upon the overall quality of this manuscript.

I am not sure why regularly stakeholders are mentioned in the article. I think it concerns 152 broiler farmers.

Response: We have edited the needed parts of the manuscript to address this.

The biggest concern about this article is the fact that it is presented as a descriptive study only in numbers of participants and percentages. In itself that is a good start, but it is not an in depth-analysis. Antibiotics is a high impact journal requiring a more sophisticated analysis, in my opinion. A second layer of analysis and suitable methods would be expected to be performed to associate the responses to underlying factors as size of farm, i.e. stocking batch and age category of farmer and farm category. Because numbers are rather small, you may make logical categories (add categories together). For instance, I would not be surprised that Industrial farms score quite different from other categories. The same with age groups.

Response: We really appreciate the reviewer for these great comments. We are committed to presenting a manuscript befitting of the standard of the journal. Hence, we re-worked the analysis by including some inferential statistics (chi-square test and binary logistic regression analysis) to address the concern. More tables have been added containing the results on the independent factors associated and influencing the knowledge, attitudes, and risk perception of the broiler grow-out farmers on antimicrobial use and resistance in the study area.

Introduction is the best part.

Response: We appreciate this comment. However, we have edited the introduction further to make it better.

Table 1 in Material and Methods 2.4 is confusing.  I think it are results. First 3 columns: which units? Maximum obtainable score? It is clear what is a satisfactory score. I guess the unsatisfactory score are just all other respondents' answers. I can imagine the Demographic characteristics (table 2) to belong to the Material and methods section.

Response: Table 1 contains the outcome variables tested as well as the total scores obtainable and obtained by the respondents after attempting the questionniare, a reason why this is presented in the materials and method section. Also, we have included a description of the satisfactory scores in the data analysis section.

I would also like to see the presented demograhic data from this studied sample to be compared with the overall situation of the sector in that State. In other words, how does the sample compare to the total broiler farming sector characteristic in this State (and Nigeria). Discussion part should preferably refer to more published research about attitudes towards AMR, like in the pig, veal calves and cattle sectors to gain a wider perspective of the results.

Response: The discussion has been improved upon to include this. We have compared our findings with available reports in respect to on other farm animals in the country.

Reviewer 2 Report

Thank you for the opportunity to review this manuscript by Oloso, O.N.O. et al. entitled “Knowledge, Attitude and Risk Perception of Broiler Grow-out Farmers on Antimicrobial Use and Resistance in Oyo State, Nigeria”.

To investigate the level of knowledge, attitude to use and risk perception of AMR of farmers in Nigerian broiler grow-out setting, the authors conducted a cross-sectional survey supported by a 5-sectional questionnaire.

To contextualize the manuscript for the reader, the introduction and background described the AMR problem and the survey type chosen. The study design as well as the data collected are clearly reported. Although I did not found major issues with the manuscript, I only have some minor issues, which help to improve the manuscript:

  • Please check and adjust the spaces along the manuscript. Especially in the lines: 76, 90, 100, 235, 339, 385 and 411.
  • The figure 1 quality should be improved.
  • Line 221: there are two points. Please correct it.
  • Line 296: there is no dot at the end of the sentence.
  • Line 391: please correct the parenthesis.
  • References: review the references following MDPI's instructions.

Author Response

Reviewer #2

To contextualize the manuscript for the reader, the introduction and background described the AMR problem and the survey type chosen. The study design as well as the data collected are clearly reported. Although I did not found major issues with the manuscript, I only have some minor issues, which help to improve the manuscript:

  • Please check and adjust the spaces along the manuscript. Especially in the lines: 76, 90, 100, 235, 339, 385 and 411.
  • The figure 1 quality should be improved.
  • Line 221: there are two points. Please correct it.
  • Line 296: there is no dot at the end of the sentence.
  • Line 391: please correct the parenthesis.
  • References: review the references following MDPI's instructions.

Response: We appreciate the reviewer for the kind comments on this work. We have attended to all the above concerns raised within the manuscript.

Reviewer 3 Report

Oloso et al., studied the knowledge attitute and risk perception of Antimicrobial use and resistance in broiler farmers. Overall the study contributes in understanding the antimicrobial usage trend in Nigerian farmers. There are some points that need to be addressed as mentioned below:

  1. It is not clear how the farmers get access to antimicrobial agents. Are they freely available over-the-counter without prescription? This point should be clearly discussed as it is a critical point that would need attention at the governmental level in controlling the inappropriate use of antibiotics.
  2. Figure 1 is poorly presented and compressed disproportionally. It should be modified to make it more clear and appealing. 
  3. Perhaps the authors can analyse the correlation between the farmers with unsatisfactory attitude to practice of antimicrobial usage (Table 1) and risk perception If I know unconscious use of AU in broiler....not be rejected (Table 5). I expect it would be correlated in these group of farmers. 
  4. There are some minor comments:
    1. line 49-51: Not clear, is it a typographical mistake?
    2. line 180: antibiotics don't kill bacteria
    3. line 191: Unnecessarily splited. "I don't fully read to understand the information on the label and prospectus before usage"
    4. line 197-200:difficult to understand, too many 'to's. Please modify into simpler sentences.
    5. line 227:Data were...2013, analysed....
    6. line 278: use of antimicrobial using antibiotic use? what does this mean?
    7. line 282: don't write very many of them. use other appropriate expression, for instance majority of them
    8. line 291: don't write vet, use the full form
    9. line 322 and 325: avoid using the word pathetic, use another way of expression, for example, surprisingly, unexpectedly.. or the like.

Author Response

Reviewer #3

Oloso et al., studied the knowledge attitute and risk perception of Antimicrobial use and resistance in broiler farmers. Overall the study contributes in understanding the antimicrobial usage trend in Nigerian farmers. There are some points that need to be addressed as mentioned below:

  1. It is not clear how the farmers get access to antimicrobial agents. Are they freely available over-the-counter without prescription? This point should be clearly discussed as it is a critical point that would need attention at the governmental level in controlling the inappropriate use of antibiotics.

Response: We have included a discussion of this crucial point raised. Thank you!

  1. Figure 1 is poorly presented and compressed disproportionally. It should be modified to make it more clear and appealing. 

Response: We have modified Figure 1 to a better resolution.

  1. Perhaps the authors can analyse the correlation between the farmers with unsatisfactory attitude to practice of antimicrobial usage (Table 1) and risk perception If I know unconscious use of AU in broiler....not be rejected (Table 5). I expect it would be correlated in these group of farmers. 

Response: We have carried out further inferential statistics to address this concern. The results have been presented within the manuscript.

  1. There are some minor comments:
    1. line 49-51: Not clear, is it a typographical mistake?
    2. line 180: antibiotics don't kill bacteria
    3. line 191: Unnecessarily splited. "I don't fully read to understand the information on the label and prospectus before usage"
    4. line 197-200:difficult to understand, too many 'to's. Please modify into simpler sentences.
    5. line 227:Data were...2013, analysed....
    6. line 278: use of antimicrobial using antibiotic use? what does this mean?
    7. line 282: don't write very many of them. use other appropriate expression, for instance majority of them
    8. line 291: don't write vet, use the full form
    9. line 322 and 325: avoid using the word pathetic, use another way of expression, for example, surprisingly, unexpectedly.. or the like.

Response: We thank the reviewer for these comments. These have been addressed.

Round 2

Reviewer 1 Report

This article has improved in set-up by adding a second layer of analysis

I still miss a paragraph in the discussion in which the main characteristics of the sample of farmers (lines 254-258) are globally compared to the the actual situation in Oyo State. Do these characteristics somewhat reflect the actual situation in the region? If such statistical data are not available in this region, then just mention this.

I am not in favour of a precision of 2 decimals for percentages listed in tables, etc.. Outcomes of such questionnaire research are not that precise in practice and can be viewed as more indicative.

Author Response

This article has improved in set-up by adding a second layer of analysis

We thank the reviewer for the positive comments on our manuscript. We have used the comments to improve upon the quality of the manuscript. The manuscript has also been check for language use and style.

I still miss a paragraph in the discussion in which the main characteristics of the sample of farmers (lines 254-258) are globally compared to the the actual situation in Oyo State. Do these characteristics somewhat reflect the actual situation in the region? If such statistical data are not available in this region, then just mention this.

Response: The data is very scarce on the specific area on data. A paragraph has been included in the discussion relating this information. A statement in the first paragraph of the Discussion has been expanded to explain it. “Our search revealed this study as the first that focused on exploring the contribution of the farmers’ broiler production setting to the awareness of AMR threat. Therefore there is scarce resources available to compare and relate the demographic of respondent in this region to other areas globally”.  

I am not in favour of a precision of 2 decimals for percentages listed in tables, etc.. Outcomes of such questionnaire research are not that precise in practice and can be viewed as more indicative.

Response: We have ensured that the data is corrected and presented in 1 decimal place.